# Animal-Based Indicators for On-Farm Welfare Assessment in Sheep

**DOI:** 10.3390/ani11102973

**Published:** 2021-10-15

**Authors:** Romane Zufferey, Adrian Minnig, Beat Thomann, Sibylle Zwygart, Nina Keil, Gertraud Schüpbach, Raymond Miserez, Patrik Zanolari, Dimitri Stucki

**Affiliations:** 1Clinic for Ruminants, Vetsuisse-Faculty, University of Bern, 3012 Bern, Switzerland; romane.zufferey@students.unibe.ch (R.Z.); adrian.minnig@students.unibe.ch (A.M.); sibylle.zwygart@vetsuisse.unibe.ch (S.Z.); patrik.zanolari@vetsuisse.unibe.ch (P.Z.); 2Vetsuisse-Faculty, Veterinary Public Health Institute, University of Bern, 3012 Bern, Switzerland; beat.thomann@vetsuisse.unibe.ch (B.T.); gertraud.schuepbach@vetsuisse.unibe.ch (G.S.); 3Centre for Proper Housing of Ruminants and Pigs, Federal Food Safety and Veterinary Office, Agroscope, 8356 Ettenhausen, Switzerland; nina.keil@agroscope.admin.ch; 4Consulting and Health Service for Small Ruminants, 3362 Niederönz, Switzerland; raymond.miserez@caprovis.ch

**Keywords:** small ruminants, sheep, welfare, animal-based, indicators, protocols, on-farm, assessment

## Abstract

**Simple Summary:**

As we keep and use sheep, we need to be able to assess their welfare and deal with welfare problems as they arise. To assess welfare, a comprehensive protocol based on valid and feasible indicators is needed. The aim of this study was to review the scientific literature and identify protocols and indicators for assessing the welfare of sheep. We identified promising protocols, well-known and established indicators, such as lameness or body condition score, as well as novel indicators that still need to be evaluated to prove their validity, such as pruritic behaviour or resting time. This review provides a starting point for the development of valid and feasible on-farm protocols using animal-based indicators to assess sheep welfare.

**Abstract:**

The value society assigns to animal welfare in agricultural productions is increasing, resulting in ever-enhancing methods to assess the well-being of farm animals. The aim of this study was to review the scientific literature to obtain an overview of the current knowledge on welfare assessments for sheep and to extract animal-based welfare indicators as well as welfare protocols with animal-based indicators. By title and abstract screening, we identified five protocols and 53 potential indicators from 55 references. Three out of the five protocols include animal-based as well as resource-based indicators. All of them were assessed as being practicable on-farm but lacking reliability. Some of the single indicators are endorsed by the literature and widely used in the field like assessment of behaviour, lameness or body condition score. Others (e.g., Faffa Malan Chart FAMACHA©, dag score or pain assessment) are regularly mentioned in the literature, but their reliability and usefulness are still subject of discussion. Several indicators, such as pruritic behaviour, eye condition, lying time or tooth loss are relatively new in the literature and still lack evidence for their validity and usefulness. This literature review serves as a starting point for the development of valid and practicable welfare protocols for sheep.

## 1. Introduction

Animal welfare has always been an issue of concern to varying degrees in our society and has evolved enormously over the years. As animals cannot express their needs directly, their welfare depends on our interest and understanding, as well as our diligence in measuring, respecting and improving the conditions for the animals we keep [1]. In order to get an impression of the live quality of stock animals, we need to be able to assess their welfare with practical and robust protocols and be able to address welfare problems as they occur [2]. To meet these expectations, it is imperative that a valid and understandable protocol, based on coherent indicators, must be developed to attest welfare.

Animal welfare indicators can be sorted in three categories: (i) indicators assessed by observation or examination of animals (animal-based); (ii) indicators that assess animal-related provisions such as housing and grazing (resource-based); or (iii) indicators that relate to farmers’ policies and management practices (management-based) [3]. Animal-based indicators of sheep welfare selected for a welfare assessment must be valid (relevant to sheep welfare), reliable (produce consistent results when performed at different time points or by different assessors) and feasible (efficient in terms of time, staff and materials) [4].

Several studies have identified the main welfare problems of sheep [5,6], and some studies have provided welfare protocols [5,7,8,9,10], or identified single welfare indicators. The aim of this review was to evaluate the possibilities to assess sheep welfare and to provide an overview of how appliccable these possibilites are considered to be on-farm. We review the scientific literature on either assessment protocols or single animal-based welfare indicators for sheep, and state their value in terms of validity (does the indicator reflect welfare?), reliability (how accurate is the indicator between observers and over time?) and/or feasibility (is the indicator considered practical in terms of time and resource consumption?). For this review, we chose to select animal-based welfare indicators since many experts consider them the most valid method to assess animal welfare. Such indicators provide a direct measurement of the welfare status of the animals and often reflect the outcome of resource inputs and management practices [11].

## 2. Materials and Methods

### 2.1. Literature Search

A search of scientific literature using assembled search terms accepted by consensus of experts was performed. Included were experimental and observational studies on sheep welfare referring to welfare assessments for adult sheep or lambs. For this purpose, five main terms (“ewe”, “lamb”, “ovine”, “sheep”, “small ruminants”,) were separately combined with twelve secondary terms out of four categories (“data”, “health”, “mortality”, “welfare”). The search was conducted in four search engines: PubMed [12], Science Direct [13], Scopus [14] and Web of Science [15]. We considered all articles published between the 1 anuary 1995 and the 4 March 2020. Any query that resulted in more than 500 articles was narrowed down with filters to reduce the number and improve the precision of the results. The used filters were set sequentially until fewer than 500 papers were listed. First, we chose filters to include only publications written in English, French or German, second, we used filters to exclude those related to human medicine (e.g., “animals”) and finally, we isolated papers specifically in the field of veterinary medicine (e.g., “veterinary sciences”). At the end, 24,675 papers were compiled. All papers were saved using the literature management software Zotero 5.0.93 (Vienna, VA, USA) [16].

### 2.2. Title and Abstract Screening

For an evaluation of the content, the articles were transferred to DistillerSR (Ottawa, ON, Canada) [17] and checked for duplicates, allowing the removal of 9606 duplicates. The remaining 15,069 articles underwent a title screening by the first author to sort out all publications that did not match the purpose of this review. We excluded publications not written in English, German or French, if were not filtered during the search-result reduction step, and publications which were clearly not related to sheep welfare. This step reduced the number of articles to 749, which were then subject to abstract screening. For the subsequent data extraction step, only peer-reviewed journal articles that contained information on animal-based indicators of sheep welfare or protocols, defined as any procedure that includes information on the measurement of sheep welfare using multiple indicators, were forwarded. After this step a total of 51 references remained in the selection. 

### 2.3. Data Extraction

The remaining 51 articles were evaluated to identify animal-based indicators for sheep welfare. We included all articles with descriptions of welfare indicators or protocols and specifically assessed the authors’ discussion of these indicators in terms of validity, reliability and/or feasibility. We excluded 17 articles, as the discussions of the respective articles did not contain sufficient information on the authors’ appraisal of the presented indicators in terms of the selected quality features. From the remaining 34 papers, 21 additional articles mentioned in the references, were included using the same procedure as described in the previous sections. The extracted indicators and protocols were registered in a Microsoft Excel v2102 (Redmond, WA, USA) [18] spread sheet. Indicators with very similar definitions, e.g., fleece condition and fleece derangement, were grouped together, resulting in 53 individual indicators that were considered relevant to this review. We classified the indicators according to the number of publications in which they were mentioned and report and discuss them in the results section according to citation frequency in descending order. Additionally, there were five articles describing procedures that included multiple indicators united in an entire animal welfare assessment protocol. These articles are reported starting with the most comprehensive protocol.

## 3. Results and Discussion

### 3.1. Welfare Assessment Protocols

Established protocols may be the easiest way to assess sheep welfare, as they cover more than one aspect of welfare using a set of different indicators. We identified five protocols from the scientific literature, which we subsequently denote after their authors or, if available, their name [5,7,8,9,10]. All protocols consist of animal-based indicators and were declared to be practicable on-farm by the respective authors.

#### 3.1.1. AWIN

Well-founded animal welfare protocols emerged from the Welfare Quality^®^ (Leystad, Netherlands) [19] project established in 2004 for cattle, pigs and poultry, but none for small ruminants. In response, the Animal Welfare Indicators (AWIN) project was established in 2011 [9,20]. The AWIN project was developed with the aim of improving animal welfare and filling a gap in the Welfare Quality^®^ project [9]. The AWIN protocol is based on a two-level approach, a prior herd-level approach and an in-depth individual-level assessment [20]. The indicators for each level are provided in Table 1. At the first level, a screening of the flock is carried out with robust and rapid animal-based indicators with no or minimal animal handling. Performing the second level assessment is recommended when there is a non-compliance with the current animal welfare legislation or if the assessment of a specific indicator results in the assessed farm belonging to the lowest 5% of the farms in the reference population. The second level consists of a more detailed and an in-depth assessment requiring restraining the animals and collecting individual data [20]. The two-level approach was chosen to reduce animal’s stress and the time needed for the assessment.

#### 3.1.2. Protocol of Napolitano 

Napolitano et al. elaborated a protocol in Italy [7]. The protocol is based on four categories derived from the Animal Needs Index [21] for cattle. These four categories focus mainly on resource-based parameters, whereas, a fifth category also includes animal-based parameters. The fifth category, which accounts for 36.6% of the total score, contains seven animal-based indicators presented in Table 1; body condition, integument alterations, animal dirtiness, hoof overgrowth, lameness, lesions and mutilations, such as de-horning and tail removal. In addition, body condition was chosen as an indicator of malnutrition and disease. According to the authors, the indicator body condition score could not be evaluated in practice because the sheep were not shorn as the farms were visited in winter. Therefore, thin and even emaciated sheep could not be identified, and body condition did not contribute to the evaluation of welfare even if the authors recommend this indicator for inclusion in a welfare protocol. Each measure results in a score depending on its prevalence in the herd. The more frequent violations for a specific indicator are observed within the herd, the lower the assigned score. The score “optimal” is given if 5% or fewer animals in the herd are observed with a violation, “good” for 10% or fewer, “medium” for either 50% or 25% or fewer, depending on the indicator, and poor for more than 50% or 25%, respectively. The separate indicator scores are then expressed as numeric values and summarized to a final score.

To evaluate the protocol in terms of feasibility and inter-observer reliability, two trained observers carried out the protocol on ten organic and ten conventional sheep farms in Southern Italy. The average number of animals per farm was 350, and at least 20% of lactating animals were recorded on each farm. No sophisticated equipment was required and the average time to complete the assessment was 85 minutes per farm. The authors identified the lack of direct measurement of internal parasites as a weakness of the protocol and recommend the inclusion of parasite egg counts to increase the validity of the scheme, although the assessment time may increase. The authors also advised good training in lesions assessment and to visit farms soon after shearing to facilitate lesion detections and body condition assessment and to increase the reliability of interventions.

Based on these findings, the protocol seems to be a practical tool for assessing the welfare of sheep on-farm. In addition, the protocol could provide farmers with recommendations on which management aspects need to be improved. However, further studies are needed to test the scheme on a larger sample size to assess its reliability [7].

#### 3.1.3. Protocol of Stubsjøen

Stubsjøen et al. [8] also proposed to assess the welfare of sheep using animal- and resource-based measurements. They adapted the animal welfare protocol established for dairy cattle, based on the Five Freedoms [12] to sheep. Sixteen animal-based, 15 resource-based and three measurements on production records (slaughter weight, carcass classification and fat class) were selected. The animal-based indicators are presented in Table 1. The protocol consists of two parts; the animal- and resource-based measurements carried out during farm visits and the analysis of production data. The assessment starts with a flock observation to detect signs of clinical disease, lameness and coughing. Then, ten randomly selected animals undergo a clinical examination. Finally, in the animal-based measurements, the animal’s behaviour is observed to assess anxiety levels and the human–animal relationship.

In relation to the indicator measuring fear, the authors used a modification of methods validated in Reference [22] to assess the ewes’ response to an unfamiliar person. In brief, the indicator counts how often a test person can walk up to and touch selected animals. To test the farmer-animal relationship, the person who interacted most with the animals was asked to enter different pens and tag randomly selected ewes in each pen. The ewe’s response was categorised into four groups, ranging from 3) “behaved calmly when approached” to 0) “attempts to escape by jumping out of the pen”. An average score was calculated for each farm. Finally, resource-based indicators such as relative humidity or temperature are measured three to 27 times, depending on the farm’s sizes, and an average is calculated. On average, three to five hours are needed to carry out the assessment and all the observations are carried out indoors. The second part, the analysis of production data, includes individual information on carcass weight, fat class and carcass classification. Regarding these three indicators, the authors could not find sufficient information on their value as indicators.

To test the protocol in relation to inter-observer reliability, two observers with clinical experience from veterinary practice and one ethologist visited 36 farms in Norway and assessed ten randomly selected animals on each farm. The assessment took place during lambing season. The observer’s agreement was excellent, except for body condition score (BCS), callus on carpus and claws. Therefore, the scoring systems for these three measures need to be more clearly defined or the observers have to be trained in more detail. Furthermore, the reliability and feasibility of the selected parameters still need to be assessed [8].

#### 3.1.4. First and Second Protocol of Munoz

Two protocols by Munoz et al., published in 2018 [5] and 2019 [10], include animal-based indicators and were developed for extensively managed sheep. The authors derived 17 indicators from a review of the relevant scientific literature (Table 1) and matched them to the five domains of welfare. Of the 17 measurements, eight were selected for their reported validity and their reliability and feasibility [5] to assess the welfare of extensively managed ewes.

The first protocol was tested on 100 randomly selected ewes from a larger flock of about 3000 breeding ewes in Victoria, Australia. Each animal was studied at three key stages: pregnancy, lactation and weaning. The ewes were kept in four groups of 25 animals. First, a group flight distance test was conducted to observe the ewe’s response to an unfamiliar human. Then, the ewes were placed in a single row and were individually examined. The indicators included in this protocol were able to detect impaired welfare and welfare risks in extensively managed systems [5] but their reliability and feasibility need further research.

The second protocol, by the same authors is an adaptation of the first one [5]. Of the eight animal-based welfare indicators, six were kept: body condition score (BCS), fleece condition, skin lesions, tail length, dag score and lameness (Table 1). In addition, the number of ewes that required further care, defined as sick or injured, was recorded.

This protocol was tested on 32 commercial sheep farms in Victoria, Australia. For the protocol animal-based indicators were considered to be the most important, but the authors state that some relevant management- and resource-based indicators, such as nutrition management or shelter provision should also be included in future assessments. According to their judgement a combination of animal-, management- and resource-based information could lead to a better understanding of potential problems for sheep welfare and how they could be avoided or minimised best. This protocol seems as well to be able to identify and assess the main sheep welfare issues as the first one but with fewer indicators [10].

### 3.2. Single Indicators

Thirty welfare indicators could be extracted from the scientific literature for which data on validity, reliability or feasibility exists in multiple articles. They are listed and discussed below from the most frequently cited to the least. Further 23 indicators were cited only once [23,24,25,26,27,28,29,30,31,32]. Thus, the amount of information is too small to judge the value of these indicators in terms of a general applicability. These indicators are listed in Table 2. Further research is needed to estimate the validity, reliablity and feasibility before these indicators can be recommended or rejected from inclusion in a general welfare assessment protocol for sheep.

#### 3.2.1. Behaviour Assessment

The most suitable stress assessment for routine on-farm checks seems to be a behavioural observation [6,33]. For example, feeding or rumination behaviours have been suggested by experts [6] as good indicators of positive conditions in sheep [30]. However, neither the method of assessment nor its reliability have been described in these articles.

Categorizing animals as “obviously sick” would allow an overall impression [23], as sheep suffering from welfare issues can be recognised through their dull, depressed demeanour [6,26]. Conducting this kind of observations, in sheep or in lambs, has proven to be feasible on-farm as the animals do not need to be gathered or handled [26,32] and showed a promising level of intra- and inter-observer reliability [24,32].

To gain a systematically assessed insight into the animal quality of life, the qualitative behavioural assessment (QBA) chose to assess how an animal demonstrated a behaviour rather than the behaviour itself. The focus of this method is to keep the whole animal perspective, and to assess observed details of posture and behaviours in the light of the entire animal’s interaction with its environment. To do so, a list of characteristics, such as content, sociable, playful or irritable, can be prepared in advance or developed by the observers themselves [34].

This assessment has been tested at many different levels and may be the most promising indicator for assessing positive emotional state in sheep, as it is considered both valid and feasible [4]. It has been used to assess sheep’s behaviour via video [35], on-farm [36,37] and during transport [38]. According to the studies, QBA has the potential to serve as a sensitive, meaningful indicator for assessing sheep welfare due to its feasibility, reliability and correlation with physiological responses. In addition to providing an overview of the animal’s behaviour, QBA appears to allow the identification and monitoring of sheep with intestinal worms and those requiring treatment [28]. The background information on the farm given to observers, did not substantially affect the relative rankings of animals on the main expressive dimensions (i.e., the pattern of interpretation), but did sensitise observers to certain aspects of the observed sheep’s expression. Therefore, in accordance with all the studies cited above, the need for good training for observers prior to the assessment was pointed out [39].

#### 3.2.2. Lameness

Lameness is a significant problem affecting young and growing lambs as well as adult ewes and rams. As any production group can be affected, the presence or absence of lameness seems to be a good indicator to include in an animal welfare protocol [6,40]. Several lameness and gait scoring systems have been developed using different categories [41,42]. Even if all scoring systems produce a fair to good level of inter- and intra-observer reliability [25,26], a binary scoring scale that rates the animal as “healthy” or “lame” appears to be the most reliable and practical method for sheep [24,43] and lambs [32]. When used on-farm, a group assessment appears to be more feasible and shows a slightly higher percentage of lameness detection [26,43]. Reference [43] suggests that the higher prevalence in group assessments originate from an increased difficulty in detecting lameness in sheep that are stressed from an inspection in isolation. In conclusion, each of the aforementioned studies concludes that this indicator is a robust and viable tool for on-farm assessment and recommend its inclusion in the animal welfare protocol. Lameness as a welfare indicator was also included in all the above discussed protocols. 

#### 3.2.3. Body Condition Score (BCS)

The BCS is a scientific measure for assessing the degree of fatness or condition of the animal using a descriptive score from 0 to 5. It is easy to learn and use and requires no equipment [44]. Even if the assessment needs handling of the sheep, the BCS shows a good on-farm acceptance and may be the most direct method of assessing persistent hunger in sheep [4]. BCS varies throughout the production cycle and knowing how BCS changes during the shepherding year allows the identification of individual animals with welfare or health problems [40]. This indicator appears to be a valid quantitative predictor of animal welfare [45] as well as a monitoring tool for selective treatment of internal parasites as part of the Five Point Check© [27]. The method shows good inter-observer reliability [25,26], which could still be improved by simplifying the scale to a fit-fat-thin score [6,26]. Because the method is based on a subjective assessment [44], the need for good training is of vital importance [32]. In short, all studies recommend the inclusion of a BCS or a fit-fat-thin assessment in animal welfare protocols. 

#### 3.2.4. Faffa Malan Chart (FAMACHA©)

The Faffa Malan Chart (FAMACHA©) system is a colour chart for the non-invasive detection of anaemia in small ruminants [46] and for targeted selective treatment of gastrointestinal parasites, as part of the Five Point Check© protocol [27]. The FAMACHA© chart shows no interrelationship with faecal egg count, but has been shown to correlate with haematocrit [47] and, therefore, seems to be a valid indicator for *Haemonchus* sp. [48] as well as adult *Fasciola hepatica* and could be used to identify sheep with high established fluke burden [49]. However, the chart shows low sensitivity in growing lambs, with an accuracy level of only 50% in identifying lambs in need of treatment [50,51] and should not be used alone to control haemonchosis in young animals [52,53]. 

In conclusion, the FAMACHA© chart correlates with haematocrit and could therefore be used as an indicator of anaemia in sheep. This method may be relevant to identify blood-feeding gastrointestinal parasites such as *Haemonchus* sp. and adult *F. hepatica*, but only in adult sheep.

#### 3.2.5. Ears Postures

Three pain assessments for sheep or lambs could be found in the literature: the Sheep Pain Facial Expression Scale (SPFES), the Sheep Grimace Scale (SGS) and the Lamb Grimace Scale (LGS). All scales assess expression in different facial areas that are rated in three categories of abnormal expression “absent”, “partially present” or “present”. The SPFES shows a high degree of accuracy in detecting suffering sheep. According to the observers, SPFES is easy to assess, and their study showed a high inter-rater reliability and high consistency. The SPFES seems to provide a reliable and effective method for assessing pain in sheep after minimal training [54]. The SGS was also shown to be a valid and reliable method for identifying distress in laboratory sheep [55]. In contrast, the LGS results should be taken with caution due to the small number of lambs (only nine) used in the study. Nevertheless, the LGS score increased significantly from before to after painful interventions had been carried out, while the score of the control lambs remained the same. These results suggest that trained human observers were able to apply the LGS and distinguish suffering lambs from control lambs [56].

Both the LGS and the SPFES consider ears that are tense and pointed backwards or downwards as a reliable sign of pain in sheep [54,56]. Ears pointed backwards could, however, as well be a sign of an uncomfortable situation or fear [57]. The SGS describes a slightly different scale with erect ears as a sign of no pain, flattened ears as moderate indication of pain, and hanging ears as severe pain [55]. Yet, when sheep are being brushed, horizontal and backward ears with only few ear posture changes seem to reflect a neutral or even positive state [31,57,58,59]. The breed characteristics may also be an important factor in interpreting ear posture, as ear posture may vary between breeds [31]. Nevertheless, changes in ear position should remain the same [54]. Because of the conflicting reports and difficulty to interpret ear postures, further research is needed to determine its usefulness as an indicator for the wellbeing of sheep.

#### 3.2.6. Eye Aperture

Eye aperture or orbital tightening has been suggested as an indicator of positive emotions as well as pain in sheep and lamb depending on the situation in which it is observed. This feature has been observed during brushing and shows that sheep seem to close their eyes while experiencing positive emotions [30,31]. Eye opening correlates well with cardiac measures and would be readily applicable on the farm using descriptive categories, such as “wide open” and “half closed” eyes [58]. However, eye opening is also a component of three pain scales, namely the SPFES, SGS and LGS under the name “orbital tightening”. All three interpret the “squeezing” of the eye or the narrowing of the eye aperture as a sign of pain [54,55,56]. The eye aperture seems to be a valid component of the pain scales, but not on itself, as it can indicate a state of well-being as well as a state of pain.

#### 3.2.7. Comfort around Humans

Sheep’s alertness to approach in the field has been recognized as a potential welfare indicator [6]. A common assessment constitutes the human approach test. This test involves observing the animal’s reactions when approached by a human. Behaviours such as escape attempts or aggression are typically expressions of fear. These reactions are seen as possible indicators of discomfort around humans [60]. It is debatable, whether the approaching human should be familiar to the sheep (which might be more relevant in terms of welfare when sheep are in daily contact with their keeper) or not familiar (which might be better standardized across farms). Another way to assess the animal’s comfort around humans might be the fear test [4]. This test is based on observing the behaviour of sheep in the presence of an unmoving human [22]. It has been used to detect fear behaviours in lambs, such as inhibition of feeding, long distance from the frightening stimulus, frequent immobilizations, and numerous high-pitch bleats [61]. Both the human approach test and the fear test performed with indoor ewes have the potential to be used for on farm welfare assessment. However, both require further work to develop the details of the methods and to assess the reliability of the test [4].

#### 3.2.8. Fleece Condition

Presence or absence of wool loss and fleece condition have been suggested by stakeholders as indicators of well-being in sheep [6,26]. Fleece condition can be a strong early indicator of the presence of aphids [62] or ectoparasites such as *Psoroptes ovis*, which can have a significant negative impact on sheep welfare [24]. These indicators appear to be more reliable and easier to assess than pruritic behavior [62]. Furthermore, group assessment via fleece condition appears to be reliable, yet further research is required to determine the optimum group size, as closer observation of individual animals may be required to identify areas of wool loss [24]. The indicator shows high inter-observer reliability at every production stage [25]. Fleece condition assessment was judged to be feasible and suitable for inclusion in sheep welfare protocols by all the studies mentioned above.

#### 3.2.9. Faecal Soiling or Dag Score

Faecal soiling may occur as a result of a complex interaction of factors, such as gastrointestinal infections [63] or high-quality spring grass [4]. The proportion of faecal soiling correlates with faecal egg counts and therefore with worm burden. The degree of faecal soiling can be assessed by scoring the animal according to the size of the region soiled around the breech; using a dag score between 0 and 5, where a score of 0 represented a clean breech region and 5 described a breech region where faeces adhered to more than two thirds of it [63]. The dag score is part of the Five Point Check© for selective treatment of internal parasites in small ruminants. South African farmers consider the dag score as understandable and useful for worm causing diarrhoea [27]. Depending on the study, its reliability varies from poor to high, but it has been recognized as rapid, non-invasive [63] and feasible [24] and should be included in animal welfare protocols [25] at least because faecal soiling is a risk factor for fly infestation and therefore remains relevant for sheep welfare [4].

#### 3.2.10. Skin Lesions

Skin lesions or wounds are considered highly valid welfare indicators as they provide a direct assessment of the presence or absence of injuries [4,6]. Large skin lesions are easily observed, but small lesions are more difficult to identify [23] and may be hindered by the presence of wool. References [25,26] suggest skin lesions to be assessed on the entire body, even turning sheep over. In fully fleeced sheep, inspection is performed by parting the fleece and by palpating the skin. Therefore, handling of the animal is required to allow an efficient examination of the animals. Reference [23] states the possibility to identify ectoparasites from lesions, as the extensive scratching and biting of infected areas may cause wounds. However, the validity and reliability of such recognizing ectoparasites through lesions are, to date, unknown. The assessment can easily be performed [4] and seems to be reliable [25,26]. Based on these results, Reference [25] recommends the inclusion of skin lesion assessment in welfare protocols for sheep [25].

#### 3.2.11. Tail Docking and Tail Length

Tail docking is considered a painful procedure [6] and risk factor for infections if the procedure is poorly performed [20]. Therefore, the tail length is a key welfare issue [6] and an indicator of preceding poor welfare [23]. In line with this conclusion, a group of experts suggested tail length as an indicator of sheep welfare [6]. This measure is feasible on-farm, where it has a good reliability and can be assessed with a binary score: 0 = tail covers the anus in males or vulva in females, 1 = tail is over-shortened [25,26]. In addition, experts suggested a management-based indicator to be more feasible by recording whether tail docking was practised, and if so, which method, analgesia and anaesthesia were used, rather than measuring the pain responses of the lambs [64]. The tail length seems to be a robust and feasible indicator to include in animal welfare protocols [25,26].

#### 3.2.12. Fleece Cleanliness

Fleece cleanliness measures the extent of soiling from external sources, such as rain, mud or dirty pens, whereas, faecal soiling should be assessed as a separate indicator (see Section 3.2.9). The fleece cleanliness seems to be a promising indicator of sheep’s environmental status that can be used in further animal welfare protocols [4,23]. It achieves a good level of inter- and intra-observer reliability [25,26]. As to how exactly fleece cleanliness is recorded, the available information is scarce. Reference [25] considers the whole body, using a 4-point visual assessment, whereas Reference [26] assessed the ventral abdomen with a 3-point visual scale. References [4,23] refer to fleece cleanliness in a more general nature. Given the few sources of research, it is not possible to adequately compare the practices regarding the ideal approach to assess cleanliness. Nevertheless, this measure is easily feasible because it does not require the animals to be gathered and handled and can be performed on undisturbed animals in their home environment [4].

#### 3.2.13. Mastitis

Mastitis may be a useful indicator of ewe welfare and health and can be assessed using a variety of methods. For example, the California Mastitis Test is considered a good diagnostic technique [65]. Another example is udder palpation. Mammary glands can be palpated to identify areas of focal or diffuse thickening, swelling, heat, pain or discomfort. They can be scored as “no evidence of mastitis”, “one gland” or “both glands affected by mastitis”. This method achieved good inter-observer reliability and is considered feasible [26]. However, Reference [25] remarks that udder examination and milk samples collection are time-consuming and labour intensive, making them less attractive for on-farm use. An alternative proposed by Reference [66] would be to use altered lamb and ewe behaviours. On one side, lambs show a preference to suckle on the unaffected gland. On the other side, ewes show an increased vocalisation and prevent their lambs from suckling more frequently when affected by mastitis. This change in normal behavioural pattern could be observed as early as 3 days after infection.

#### 3.2.14. Pruritic Behaviour

Self-traumatising behaviours such as scratching and rubbing appear to be useful observations for assessing welfare in sheep infested with ectoparasites such as *Psoroptes ovis* [67] as well as *Bovicola ovis* [62]. The time sheep spend rubbing themselves correlates positively with the total lesion area and the number and age of lesions. The amount of rubbing behaviour increased with age and lesion size. However, larger lesions were associated with a decrease in the frequency of standing-up attempts followed by a rubbing attempt. This suggests that other factors associated with lesion development may affect rubbing behaviour. These factors include increased pain and discomfort, which may also interfere with the lying behaviour of infested sheep [68].

#### 3.2.15. Diarrhoea Score

The diarrhoea score (DISCO) is used to describe the sheep faeces with a score of 1 corresponding to normal sheep faeces in pellets, 2 for “soft” faeces (similar to cow pat) and 3 for diarrhoea (semi-liquid faeces) [69]. Presence of diarrhoea seems to be a valid indicator with a significant relationship to the intensity of intestinal parasite infestation in lambs. This score allowed to correctly identify 80% of the animals in need of treatment [50]. The DISCO score was lower in healthy animals or those infected only with nematodes than in sheep infected with cestodes. It also correlated with the number of cestode but not nematode eggs per gram of faeces (EPG) [48]. According to Ref. [47], DISCO should not be used to detect early infection with *H. contortus* as it does not reflect the intensity of infection nor is it consistent with faecal egg counts.

#### 3.2.16. Weight Gain

Reduced weight gain can be associated with intestinal parasite infections. According to Reference [53] daily weight gain in lambs can be effectively used to identify lambs in need of treatment. In contrast, in Reference [50], reduced weight gain is described as not useful and without association to any other pathophysiological indicator relevant to the diagnosis of intestinal parasites. Reference [47] questioned the accuracy of weight gain reduction as it does not correlate with faecal egg count and cannot reflect the intensity of *H. contortus* infection in sheep. This measure needs further research to clarify its usefulness.

#### 3.2.17. Rumen Fill

A panel of experts identified rumen or abdominal fill as an animal-based measure of access to feed [6]. It was scored on-farm using a simple binary scale: 0 if the animal’s left-hand side was not sunken/or was convex between the hip bone and the ribs and 1 if the animals’ left-hand side was deeply sunken. The results showed a poor reliability, probably due to the difficulties to assess the rumen fill on sheep with a lot of fleece [25]. For lambs, the same indicator showed good inter-observer reliability, but due to the close observation required, 96% of lambs kept outdoors could not be scored [32]. Therefore, depending on the housing conditions the results from this indicator should be interpreted with caution.

#### 3.2.18. Excessive Panting

Excessive panting has been identified as an animal-based, non-invasive and feasible indicator for use under farm conditions [6] to assess thermal comfort [4]. Excessive panting is defined as a rapid breathing with abdominal effort, with or without rasping noise or open-mouthed stance. This indicator could be assessed without gathering the sheep, making it easily feasible. Yet, the respective study could not investigate its reliability, as no sheep were showing this behaviour [24]. Therefore, the relevance of such an indicator is debatable, and a validation should occur with herds with known suboptimal thermal comfort. Excessive panting is a specific indicator for heat stress when measured in undisturbed animals. Under other conditions, increased respiration rate may be an indicator of distress [4]. In conclusion, the reliability of excessive panting as a welfare indicator still needs to be tested on-farm.

#### 3.2.19. Eye Condition

Eye condition or abnormality has been suggested by a group of experts as health and welfare indicator for sheep [6]. An abnormal eye condition was deemed to be present if any one of the following signs was observed—blepharospasm, corneal opacity, abnormal ocular discharge, lacrimation with tear-staining of skin, conjunctivitis, or entropion. After an on-farm test and although the sheep had to be restrained for the evaluation, the assessment of eye condition was declared feasible. However, due to the small number of sheep involved, reliability could not be assessed [26]. Reference [32] uses the same indicator for lamb, but because of the close observation required for assessment, 96% of the lambs kept outdoors could not be assessed. Nevertheless, eye condition showed an excellent level of inter-observer reliability as well as a high sensitivity and specificity. Abnormalities were clearly identified. Therefore, the authors suggest that eye condition is a highly relevant indicator and should be included in future lamb health and welfare inspection tools. Eye condition seems to have an excellent level of sensitivity and specificity but needs to be tested on a larger sample size.

#### 3.2.20. Vocalisation

Sheep vocalise during social isolation, depending on breed and age class [29] and remained silent while being brushed. Considering that brushing is perceived as a positive stimulus, vocalisation could be an indicator of negative welfare [31]. Vocalisations have been shown to be associated with negative emotional reactions and have a strong correlation with the sheep activity levels [70], which could make this indicator a good predictor of an active sheep reaction to an anxious situation. Although it needs further standardisation and validation, vocalisation would be easy to assess and seems to be a valid measure of animal welfare [29].

#### 3.2.21. Mouth Features

Mouth features are used as pain indicator and are included in three different pain scales discussed before. In the SGS, pain is assessed using three levels: (i) “closed mouth” indicating absence of pain, (ii) “puckered lips” indicating moderate pain and (iii) “flehming” representing the higher level of pain. The validity and reliability of mouth characteristics were not assessed separately from the other indicators of the SGS, including orbital tension and ear and head position [55]. In the SPFES and the LGS, the indicator is defined as flattened and tight lips with straight or slightly ventrally rotated corners. The mouth features alone do not appear to be reliable as an indicator of pain due to the low observers agreement [54,56], but may be useful as part of the various pain scores.

#### 3.2.22. Cheek Flattening

Similar to mouth characteristics, cheek tightening or flattening is included as an indicator of pain in the SPEFS as well as in the LGS. For the SPFES, Reference [54] defines cheek flattening as a more convex expression of the cheek in the region of the masseter muscle and zygomatic arch and scaled this characteristic as absent, partially present or present. Cheek tightening appeared to be relatively easy to score and showed a high inter-class correlation of 82%. Reference [56] characterises less bulging cheek area or, in obvious cases, a hollowed cheek as indicators of lambs in pain. According to their observers, cheek flattening was a difficult feature to assess due to differences in camera angle or lighting. This characteristic also had a low inter-observer reliability, suggesting that this action unit contributed little to the pain assessment and therefore could be excluded from the LGS. Therefore, cheek tightening may be a useful indicator within the SPFES to assess pain in sheep, but not in lambs.

#### 3.2.23. Nasal Features

The last facial expression included in the SPFES and the LGS are nasal features. According to both, References [54,56], sheep or lambs in pain showed a tightening nose with a decrease in nostrils, resulting in a “V” shape. Although they agree on the validity of this indicator, the results of their study diverged on the reliability. McLennan et al. found that the nose features did not correlate strongly with the other areas of the face and that this indicator was less reliable between scorers than the other measures of the SPFES [54]. In contrast, Guesgen et al. showed a good inter-observer reliability for nose features in lambs. However, they pointed out that restraining lambs affected their facial expression and influenced the measure of that feature [56]. These differing opinions suggest that this indicator should be interpreted cautiously and needs to be confirmed in future studies.

#### 3.2.24. Hoof Overgrowth

Hoof overgrowth has been cited as an indicator of sheep welfare to assess ease of movement. However, currently, there are no studies, directly linking reduced movement to an increase in hoof overgrowth. This measure depends on other factors, such as the frequency with which hooves are trimmed and the ability of the animal to move if it suffers from lameness [4]. A recent study evaluated hoof overgrowth in terms of ease of application, but found poor reliability and low feasibility, likely due to the difficulty of the assessment. According to the observers, assessing the hoof overgrowth was time-consuming and not easy to do, because ewes would not stand still. The authors suggest to use broader measures, such as lameness scoring (see Section 3.2.2), which may be more relevant [25].

#### 3.2.25. Nasal Discharge

Nasal discharge has been suggested by a group of experts as a non-invasive and practicable animal-based indicator for use under farm conditions [6]. The measure is part of the Five Point Check© protocol developed for the selective treatment of internal parasites in small ruminants. In this protocol nasal discharge serves as an indicator of nasal bots such as nasal botfly or lungworms. Note that nasal discharge can also be an indicator of pneumonia or other diseases [27].

#### 3.2.26. Tail Features

Tail wagging and raised up tail are controversially discussed indicators. On one hand, because they are rarely observed in sheep, especially with tail docking, and on the other hand, because scientists do not agree on their meaning. Lambs raise and wag their tail while suckling and being brushed. Assuming that both are positive stimuli for sheep, they may be important indicators of positive states in sheep [30]. However, lambs seem to show raised tails during separation with their ewe, which might indicate that this behaviour occurs during intense negative emotional states. According to this contradictory information, the raised tail may be shown during a negative or positive emotional state, which would render this indicator useless for discriminating emotional valence [59].

#### 3.2.27. Lying Time

The assessment of lying time for individuals was proposed to measure either the comfort of the resting places or an infestation with ectoparasites (e.g., *Psoroptes ovis*). With the aim of measuring the comfort of the resting areas, Reference [4] concludes, based on the available literature, that the measure was difficult to apply in the field. The study authors mentioned that the lying synchrony of the sheep would provide sufficient information in a simpler way. The possibility of all sheep lying at the same time can be easily assessed without disturbing the animals. However, the reliability of sheep lying synchrony has not been assessed yet. The lying time may as well be an indicator of ectoparasites as sheep infested with *P. ovis* spend less time lying down at the expense of rubbing time. The development of lesions (e.g., secondary bacterial infections) may also influence lying behaviour [68].

#### 3.2.28. Shivering

Shivering is known to be a sign of cold, which would make it a potential welfare indicator for thermal comfort. Two studies have tried to use it in both sheep and lambs and came to the same conclusions: shivering had a very low prevalence and showed a low level of inter-observer reliability, possibly due to the presence of fleece, which makes it difficult to assess. Authors of both studies considered this measure unfeasible for sheep [4,32].

#### 3.2.29. Rectal Temperature

Rectal temperature is commonly used in clinical examinations and provides useful information about the animal’s health status. This measure has been proposed as an indicator for positive welfare in sheep, but has been discarded due to the lack of significant matches with positive states [31]. Handling is required to take the rectal temperature of a sheep, which could cause stress and stress-induced hyperthermia, which could ultimately bias the results. Moreover, the invasive nature of this measure may compromise biosecurity [4]. In conclusion no study supports the use of rectal temperature as a welfare indicator for sheep.

#### 3.2.30. Tooth Loss

Assessing tooth loss or dental abnormality could give an indication of the sheep’s ability to feed and could allow animals at risk to be identified earlier. Even if the assessment requires handling of sheep, this procedure is quick, simple and frequently performed on-farm, suggesting good feasibility. The reliability of this measure has not been tested [4] but the assessment of tooth loss or dental abnormalities was found to be feasible [26].

## 4. Conclusions

The aim of this study was to review the scientific literature published from January 1995 to March 2020 to obtain an overview of the articles available linked to sheep’s welfare and to extract animal-based welfare indicators as well as already established welfare protocols. For this review, a total of five protocols and 53 indicators were identified. All the protocols include animal-based indicators validated in the literature and seem feasible on-farm, that is, they need limited resources, effort and can be applied with little disturbance for the animals. However, all of them have yet to be tested on a larger scale and bigger sample sizes to be able to affirm their reliability for providing a consistent and truthful reproduction of the status of sheep welfare.

For individual indicators, the amount of data is greater than for entire protocols. This is owed to the fact, that most protocols relied on expert and stakeholder opinion to determine the included indicators. This practice to determine indicators through expert and stakeholder opinion in turn directly resulted from the high variation in the availability of research for the different indicators. Some indicators, such as behaviour assessment, lameness, BCS, fleece condition or skin lesions are frequently addressed in the literature and have acquired a status to be useful indicators for measuring sheep welfare. Others such as FAMACHA©, dag score, DISCO or the various pain assessments and their components are regularly mentioned in the literature, but opinions differ on their validity or feasibility. Finally, some of the indicators mentioned in this review, such as pruritic behaviour, eye condition, lying time or tooth loss, are relatively new and seem feasible, but their validity and repeatability has not yet been assessed in-depth. It may be possible to derive some priming information for these indicators from established indicators in other ruminants. Rumen fill and rumination behavior for example is rather well studied in cattle, but less in other ruminants. Although a direct comparison between sheep and cattle is not possible, research to establish welfare indicators may profit from prior knowledge as to which parameters to look for and which methodologies may be practical.

In our search terms we also specifically included terms for data-based indicators. Nevertheless, our literature search found no studies explicitly investigating data-based indicators, and only one assessment protocol includes three measurements on production records, yet, without clear results or a discussion on their quality. Given the increasing efforts for simplified welfare assessments, more research should be directed towards identifying useful, reliable and feasible methods to indicate the status of animal welfare from the ever growing stack of available data. For the time being, this literature review should serve as a starting point for further development of comprehensive, valid and practicable on-farm welfare protocols for sheep, which could also be used to validate future implementations of data-based indicators.

## Figures and Tables

**Table 1 animals-11-02973-t001:** List of animal-based sheep welfare indicators included in the different assessment protocols. For each indicator, the protocols in which the respective indicator is included in is given (marked by an X), sorted by the amount of protocols that include the indicator. For the AWIN protocol the inclusion in the first level (heard assessment) and second level (detailed individual assessment) is stated. Similar indicators, such as skin lesions, integument condition and skin irritation, are summarized as single row, although they may be evaluated as distinct indicators within a protocol.

Indicator	Munoz [5,10]		Napolitano [7]	Stubsjøen [8]	AWIN [9]	
	2018	2019			First Level	Second Level
Lameness/Gait score	X	X	X	X	X	X
Body condition score	X	X	X	X		X
Fleece cleanliness	X		X	X	X	X
Faecal soiling/Diarrhoea	X	X		X	X	X
Tail length/Mutiliations	X	X	X		X	X
Skin lesions/Integument condition/Skin irritation	X	X	X	X		X
Fleece quality/Fleece condition	X	X			X	X
Familiar human approach/Fear/Flight distance	X	X		X	X	
Mastitis or other udder problems	X	X		X		X
Hoof overgrowth/Hoof condition	X		X			X
Panting	X				X	
Social withdrawal	X				X	
Stereotypy, excessive itching	X				X	
Occular discharge/Eye abnormalities				X		X
Respiratory quality/Coughing				X		X
Lamb mortality					X	
Water availability					X	
Access to shelter					X	
Stocking density					X	
Mucosa color						X
Animal appears sick				X		
Swollen joints				X		
Callus on carpus				X		
Nasal discharge				X		
Ear tag torn out				X		
Rumen fill	X					
Aggression	X					
Qualitative behaviour assessment	X					

**Table 2 animals-11-02973-t002:** Animal-based welfare indicators in need of further research.

Domain	Indicator	Reference
Pain	Castration	[23]
	Ear notching	[23]
Health/Disease	Body weight	[6]
	Coughing	[24]
	Foot-wall integrity	[25]
	In-growing horns	[6]
	Joint swelling	[26]
	Myiasis	[26]
	Urolithiasis	[6]
	Mandible oedema	[27]
	Walking speed	[28]
Fear/Distress	Activity level	[29]
	Separation from the flock	[6]
Environment	Skin-pinch	[4]
	Aggression between members	[4]
Positive welfare	Body posture/head orientation changes	[30]
	Nasal/withers temperature	[31]
Lambs	Excessive salivation	[32]
	Response to stimulation	[32]
	Standing ability	[32]
	Tucked-up posture	[32]

## Data Availability

No new data were created or analyzed in this study. Data sharing is not applicable to this article.

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
