# Peer review of "Animal-Based Indicators for On-Farm Welfare Assessment in Sheep"

_animals, 2021, doi:10.3390/ani11102973_

Round 1

Reviewer 1 Report

Indicators are often associated with the use of formulas to calculate them. Were there any indicators in the analyzed literature that could be determined on the basis of calculations?

In the References section, in item 1 there is only "[" symbol. Is there not a complete bibliographic description on this point?

I think that in the article the use of surnames can be significantly reduced and instead use other markings, e.g. instead of writing "Phythian et al. [44] ..." you can write "Ref. [44]…” (line: 292). Instead of writing: According to Chylinski et al., you can write: According to [48]…

In subsection 3.2.3 on Body Condition Score (BCS), the authors wrote "It is easy to learn and use and requires no equipment" (lines: 300-301) and "Because the method is based on a subjective assessment [45], the need for good training is of vital importance ”(lines: 309-310). In the context of this information, it could be added that the methods of determining BCS using the BCS camera are known in practice, although so far widespread mainly in the assessment of dairy cattle (the article that can be cited: Association between body condition and production parameters of dairy cows in the experiment with use of BCS camera). Thanks to the BCS camera, greater accuracy of the assessment of the BCS index is achieved and the subjectivity of the assessment is reduced.

I think that in subsection 3.2.10. Skin Lesions it would be useful to give more details on where in sheep injuries are assessed. There are mainly two places on the rear legs, i.e. the tarsal joint and the tuber calcis. For more information, see the article: Hock lesions and free-stall design, which can be cited in this section. Alternatively, you can mention the damage (injuries) on the front legs, although in my practice with dairy cattle I dealt with the measurement of damage mainly to the rear legs.

I think that in subsection 3.2.11. Tail Docking and Tail Length would be worth giving more information on what is legitimate tail docking in lambs. I know this justification and the discussion on this topic in the case of heifers (dairy cattle), because I participated in it indirectly, so I know that it is worth writing a few sentences about it in the case of lambs. In this section, one could also mention more details about the method of docking the tail.

Does subsection 3.2.12. Fleece Cleanliness taken into account the cleanliness of the outer part of the fleece? In the case of the outer part of the fleece, elaborate on the question of the place of soiling, or, for example, the percentage of soiled area in relation to the entire body surface. When writing about the purity of the fleece, did the Authors take into account possible internal contamination, including insects? In general, the concept of "fleece purity" has been described in the article so generally that it is not known what specific elements related to the purity were included.

It is true that the authors wrote that the rumination behaviours (line 255) has not been fully described in the world literature so far, nevertheless one could mention attempts to look for analogies in studies with another group of ruminants, i.e. dairy cattle.

In my opinion, no names of authors (Stubsjøen et al. are listed in line 639) in Conclusion. Instead, for example, you could write: ... an assessment protocol developed by a research team from Norway ...

Articles in References should be described in accordance with the editor's requirements. This part of the article is the least developed and requires careful editing. For example, if in paper [55] the authors mention the postal address of the publisher of the Applied Animal Behavior Science journal, then there is probably something wrong with the final review of the material. The same remark applies to many of the other materials in References.

Author Response

Replies to Reviewer 1:

We are greatly thankful for the reviewer's constructive and helpful comments. The thorough assessment by the reviewer has revealed valid points for improvement, which we have implemented into our manuscript. Some points raised by the reviewer would have, in our opinion, led to an extension of the review far beyond the intended aim. We have addressed these points in our responses and explained our rationale for keeping the respective sections as they are.

1) Indicators are often associated with the use of formulas to calculate them. Were there any indicators in the analyzed literature that could be determined on the basis of calculations?

Response: We found only a single instance where indicators were based on a calculation: The protocol by Napolitano et al. determines the assigned scores based on the calculated herd prevalence of the respective indicator trait. All other indicators covered by the literature assigned scores based on descriptive features (such as lameness severity). As only a single instance for calculations provides no basis for comparison of the chosen thresholds or the underlying formulas, we decided not to discuss this issue further in our review.

2) In the References section, in item 1 there is only "[" symbol. Is there not a complete bibliographic description on this point?

Response: We thank the reviewer for pointing this out. The respective information was in a field from the citation software, which must have caused problems during submission. The reference is now correctly entered as text.

3) I think that in the article the use of surnames can be significantly reduced and instead use other markings, e.g. instead of writing "Phythian et al. [44] ..." you can write "Ref. [44]…” (line: 292). Instead of writing: According to Chylinski et al., you can write: According to [48]…

Response: We agree with the reviewer that the many "et al." can be cumbersome to read and that the study authors' names do not contribute significantly to the understanding of the text. We have therefore adjusted the text according to the suggestion of the reviewer in various places.

4) In subsection 3.2.3 on Body Condition Score (BCS), the authors wrote "It is easy to learn and use and requires no equipment" (lines: 300-301) and "Because the method is based on a subjective assessment [45], the need for good training is of vital importance ”(lines: 309-310). In the context of this information, it could be added that the methods of determining BCS using the BCS camera are known in practice, although so far widespread mainly in the assessment of dairy cattle (the article that can be cited: Association between body condition and production parameters of dairy cows in the experiment with use of BCS camera). Thanks to the BCS camera, greater accuracy of the assessment of the BCS index is achieved and the subjectivity of the assessment is reduced.

Response: We excluded any potential but hypothetical technological improvements to the indicators on purpose, as the aim of the review was to highlight the possibilities based on the current knowledge and methodological possibilities. In addition, we are sceptical, if a camera solution would indeed be more precise, as the fleece could drastically influence an automatic recognition of the correct body shape. Hence, a more thorough discussion of such technological advancements would be appropriate, which, however, is beyond the scope of this review.

5) I think that in subsection 3.2.10. Skin Lesions it would be useful to give more details on where in sheep injuries are assessed. There are mainly two places on the rear legs, i.e. the tarsal joint and the tuber calcis. For more information, see the article: Hock lesions and free-stall design, which can be cited in this section. Alternatively, you can mention the damage (injuries) on the front legs, although in my practice with dairy cattle I dealt with the measurement of damage mainly to the rear legs.

Response: For the references for which the available information was available, we have added a statement that skin lesions were assessed on the entire body [lines: 419-425]. As the size and weight difference between cattle and sheep is substantial, the acting forces and, therefore, causes that result in injuries likely differ. Therefore, a comparison between leg-based lesion assessments in cattle and the assessment of lesions on the entire body would likely not add much information to the review. We have, however, extended the Conclusion section in response to point 8), for which we now state the possibility to gain further insight for the development of novel indicators by using information from other ruminants.

6) I think that in subsection 3.2.11. Tail Docking and Tail Length would be worth giving more information on what is legitimate tail docking in lambs. I know this justification and the discussion on this topic in the case of heifers (dairy cattle), because I participated in it indirectly, so I know that it is worth writing a few sentences about it in the case of lambs. In this section, one could also mention more details about the method of docking the tail.

Response: We agree with the reviewer on the importance of this topic. Yet, tail docking practices and legislations vary from country to country. Therefore, a thorough discussion on the definition of tail docking and the resulting implications for animal welfare is beyond the scope of this review.

7) Does subsection 3.2.12. Fleece Cleanliness taken into account the cleanliness of the outer part of the fleece? In the case of the outer part of the fleece, elaborate on the question of the place of soiling, or, for example, the percentage of soiled area in relation to the entire body surface. When writing about the purity of the fleece, did the Authors take into account possible internal contamination, including insects? In general, the concept of "fleece purity" has been described in the article so generally that it is not known what specific elements related to the purity were included.

Response: We agree with the reviewer that information on fleece cleanliness was scarce. We have added more information on how the referenced articles scored fleece cleanliness (only two references stated this information). We also extended the discussion of this indicator to indicate more clearly, that the available information is too little for a thorough comparison of the implications of the scoring method on the quality of the welfare assessment [lines: 442-444 and 446-451]. Contamination from insects within the fleece appears not to be covered by these assessments, as they are intended for quick herd-level assessments. However, harmful insects are covered by indicators assessing the presence of ectoparasites.

8) It is true that the authors wrote that the rumination behaviours (line 255) has not been fully described in the world literature so far, nevertheless one could mention attempts to look for analogies in studies with another group of ruminants, i.e. dairy cattle.

Response: In our opinion, replacing missing information on sheep indicators with knowledge from other ruminants does not contribute to the aim of our work, as the needed research (and therefore revieweable literature) on validity, reliability and feasibility for sheep is not available. We therefore only draw parallels to other ruminants when the respective information is also available for sheep. In order to take the reviewer's comment into account, we have added the suggested statement to our Conclusion section, where it applies more generally than in the specific section on rumination behavior [lines: 664-670].

9) In my opinion, no names of authors (Stubsjøen et al. are listed in line 639) in Conclusion. Instead, for example, you could write: ... an assessment protocol developed by a research team from Norway ...

Response: We have removed the names from the Conclusion section and in various other places in the manuscript as suggested by the reviewer.

10) Articles in References should be described in accordance with the editor's requirements. This part of the article is the least developed and requires careful editing. For example, if in paper [55] the authors mention the postal address of the publisher of the Applied Animal Behavior Science journal, then there is probably something wrong with the final review of the material. The same remark applies to many of the other materials in References.

Response: We thank the reviewer for the thorough investigation of our entire review. We have adjusted the entire reference section to a uniform format, compatible with the MDPI recommendations.

Reviewer 2 Report

The authors have attempted to describe welfare indicators by reviewing the relevant literature.
They have been successful to some extent only. In fact, many significant papers have been omitted. Among these the extensive works performed in Scotland during the last 20-25 years. That is a particular omission, but I am sure that it was done by oversight, rather than on purpose. The authors should revisit the relevant literature and make these appropriate additions, given that, in fact, the first studies into animal welfare of sheep were performed in institutions based in that country.
Opinion: revaluation after significant improvement as indicated.

1. The manuscript is a review article, therefore obviously and by definition it is not original, exactly because it is a review. A review means that the authors present and discuss the work of other people and do not present new findings, therefore by definition the article cannot be original. 2. The authors discuss consistently all the articles that they present in their manuscript. However, they did not present all the articles available in the international literature.

Author Response

Replies to Reviewer 2:

We thank the reviewer for their comments and thorough assessment of the literature included in our review. Based on the reviewer's comments we have performed another literature search in order to find the specific literature. Given the information provided by the reviewer, we identified some articles that did not appear in our previous literature search, which we evaluated for their contribution to the aim of our review.

The authors have attempted to describe welfare indicators by reviewing the relevant literature.

1) They have been successful to some extent only. In fact, many significant papers have been omitted. Among these the extensive works performed in Scotland during the last 20-25 years. That is a particular omission, but I am sure that it was done by oversight, rather than on purpose. The authors should revisit the relevant literature and make these appropriate additions, given that, in fact, the first studies into animal welfare of sheep were performed in institutions based in that country.

Response: We thank the reviewer for contributing their expertise on literature considered for this review. We have screened again the literature from our original literature search specifically for studies performed in Scotland. In this process, we could confirm again, that the respective articles were correctly excluded because the connection to sheep welfare assessment through animal-based indicators was missing. The reasons for exclusion were either a too high specificity (genetic risk factors, molecular processes, single diseases) or sheep were not directly targeted by the research.

We have additionally performed a novel literature search, using the stated pairs of search terms and additionally added the search term "Scotland". This search yielded eight articles that were considered for a abstract and full-text screening (the title screening excluded all articles that obviously did not match the aim of our review). The eight articles are listed below after point 3). Subsequently judging these articles by their content, we could not identify any articles that would substantially add to the aim of our review. With our review we intend to summarize the knowledge on available indicators and their current applicability on-farm. It is not our goal, to evaluate literature on sheep welfare in general, but only in the context of established welfare indicators. We have clarified the aims of the review in our manuscript [lines: 61-63].

2) The manuscript is a review article, therefore obviously and by definition it is not original, exactly because it is a review. A review means that the authors present and discuss the work of other people and do not present new findings, therefore by definition the article cannot be original.

Response: We do not see the source for this comment, as we do not use words like original or novel in our manuscript to refer to our work. We will , however, check this regard with the editor to make sure that our work is correctly labeled.

3) The authors discuss consistently all the articles that they present in their manuscript. However, they did not present all the articles available in the international literature.

Response: See our response above to point 1). After an additional search for relevant literature based in Scotland, the following eight articles were considered, but ultimately excluded:

[1] Abu-Serriah, M, A M Nolan, and S Dolan. «Pain assessment following experimental maxillofacial surgical procedure in sheep». LABORATORY ANIMALS 41, no 3 (2007): 345-52. https://doi.org/10.1258/002367707781282794.

Assessment: This article is too specific for our review. The authors state: “The aim of this study was to investigate the severity and duration of postoperative pain and hyperalgesia in sheep undergoing mandibular reconstructive surgery”. As the article focuses on only a specific type of surgery, the information would not be applicable to a general indicator for welfare assessment.

[2] Best, C M., J Roden, A Z Pyatt, M Behnke, and K Phillips. «Uptake of the Lameness Five-Point Plan and Its Association with Farmer-Reported Lameness Prevalence: A Cross-Sectional Study of 532 UK Sheep Farmers.» Preventive Veterinary Medicine 181 (2020): 105064. https://doi.org/10.1016/j.prevetmed.2020.105064.

Assessment: This article is too specific for our review. The authors state: “The aims of this research were to determine the uptake of a national strategy to reduce lameness in the UK flock, known as the Five-Point Plan (5 P P); explore the association between footrot vaccination (Footvax®) use and 5 P P adoption”. The article focuses on a treatment plan, not a welfare assessment.

[3] Coghlan, A. «One Small Step for a Sheep.» New Scientist (1971) 153, no 2071 (1997):4.

Assessment: This article appeared in our original search, but was excluded as it focuses on the cloning process of Dolly and not welfare assessments.

[4] Fleming, P A, S L Wickham, C A Stockman, E Verbeek, L Matthews, and F Wemelsfelder. «The Sensitivity of QBA Assessments of Sheep Behavioural Expression to Variations in Visual or Verbal Information Provided to Observers.» Animal : An International Journal of Animal Bioscience 9, no 5 (2015): 878-887. https://doi.org/10.1017/S1751731114003164.

Assessment: This article appeared in our original search and is included in our review.

[5] Futro, A, K MasÅ‚owska, and C M Dwyer. «Ewes Direct Most Maternal Attention towards Lambs That Show the Greatest Pain-Related Behavioural Responses.» PloS One 10, no 7 (2015): e0134024. https://doi.org/10.1371/journal.pone.0134024.

Assessment: This article does present some behavioural changes in castrated lambs and their ewes. However, the article does not present an actual indicator and does not contain information on validity, reliability or feasibility of such an indicator.

[6] Jack, C, E Hotchkiss, N D Sargison, L Toma, C Milne, and D J Bartley. «A Quantitative Analysis of Attitudes and Behaviours Concerning Sustainable Parasite Control Practices from Scottish Sheep Farmers.» Preventive Veterinary Medicine 139, no Pt B (1 avril 2017): 134-145. https://doi.org/10.1016/j.prevetmed.2017.01.018.

Assessment: This article focuses on the acceptance of control practices among farmers, not welfare assessments or their applicability.

[7] Richmond, S E, F Wemelsfelder, I Beltran de Heredia, R Ruiz, E Canali, and C M Dwyer. «Evaluation of Animal-Based Indicators to Be Used in a Welfare Assessment Protocol for Sheep.» Frontiers in Veterinary Science 4 (2017): 210. https://doi.org/10.3389/fvets.2017.00210.

Assessment: This article appeared in our original search and is included in our review.

[8] Scott, P R, N D Sargison, and D J Wilson. «The Potential for Improving Welfare Standards and Productivity in United Kingdom Sheep Flocks Using Veterinary Flock Health Plans. » Veterinary Journal (London, England : 1997) 173, no 3 (mai 2007): 522-531. https://doi.org/10.1016/j.tvjl.2006.02.007.

Assessment: This article focuses on the improvement of management practices and does not contain welfare indicators or information on their applicability.

Round 2

Reviewer 2 Report

The authors have produced a fantastic article.
This is an excellent piece of work and it will make a very successful paper.

Overall merit 25/25

Only some language corrections are needed before acceptance.